# Ensemble Coding of Crowd with Cross-Category Facial Expressions

**DOI:** 10.3390/bs14060508

**Published:** 2024-06-19

**Authors:** Zhi Yang, Yifan Wu, Shuaicheng Liu, Lili Zhao, Cong Fan, Weiqi He

**Affiliations:** 1Research Center of Brain and Cognitive Neuroscience, Liaoning Normal University, Dalian 116029, China; yzlnnu@163.com (Z.Y.); wuyifan55555@163.com (Y.W.); lazarusliu@163.com (S.L.); 18883947831@163.com (L.Z.); weiqi79920686@sina.com (W.H.); 2Key Laboratory of Brain and Cognitive Neuroscience, Dalian 116029, China

**Keywords:** facial expression, categorical perception, ensemble coding, categorical boundary, summary statistics

## Abstract

Ensemble coding allows observers to form an average to represent a set of elements. However, it is unclear whether observers can extract an average from a cross-category set. Previous investigations on this issue using low-level stimuli yielded contradictory results. The current study addressed this issue by presenting high-level stimuli (i.e., a crowd of facial expressions) simultaneously (Experiment 1) or sequentially (Experiment 2), and asked participants to complete a member judgment task. The results showed that participants could extract average information from a group of cross-category facial expressions with a short perceptual distance. These findings demonstrate cross-category ensemble coding of high-level stimuli, contributing to the understanding of ensemble coding and providing inspiration for future research.

## 1. Introduction

We live in a complex world filled with abundant and detailed information. However, we can accurately perceive our surroundings at a glance. Previous studies attribute the process of perceiving multiple items presented simultaneously or sequentially to ensemble coding, in which observers extract summary statistics (e.g., mean, variance, the shape of a distribution); the common and intuitive way to accomplish this is averaging from a group of items [1,2,3]. Ensemble coding is applicable to a variety of visual dimensions ranging from low-level properties (e.g., size [4,5,6], orientation [7,8], and hue [9,10]) to high-level properties (e.g., facial expression [11,12,13], biological motion [14], and even economic value [15]).

In prior work on ensemble coding, the members of the sets were frequently derived from a continuum that changed continuously along a feature dimension from one extreme to another, and were selected from a continuum around the mean. Before each trial, the value of the mean was determined at random from the continuum’s range. The continuum may be, for example, an object’s size increasing from small to large [4,5,6], orientation changing from left tilt to right tilt [7,8], facial expression unfolding from low intensity to high intensity (e.g., [11,13,16,17]), and economic value rising from low to high [15]. These studies suggest that observers can extract an average representation from a set. Specifically, observers can correctly discriminate a probe stimulus from the average representation of a set in the mean discrimination task or they tend to recognize an unseen average representation as a member of the set that was previously presented in the member identification task. Beyond changes in quantity, various types of stimuli employed as members of a set in studies on ensemble coding were chosen from a continuum in terms of quality, such as facial expressions transitioning from happy to sad [13,17] or to angry [18,19,20]. It is possible that the set contained members from distinct categories in certain dimensions (i.e., a cross-category set) for stimuli selected from a cross-category continuum. These investigations also supported that observers can extract average representations from a set. Nevertheless, they did not systematically manipulate the category relationships of the members within a crowd, but combined cross-category and within-category (i.e., the set contained members from the same categories in certain dimensions) sets. As a result, these studies cannot determine whether the ensemble coding discovered was derived solely from within-category sets or involved a contribution from cross-category sets. Given that sets encountered in the real world often comprise different categories along certain dimensions, such as a field of flowers with varying colours, it is necessary to investigate whether observers can process the cross-category sets.

Previous investigations on this issue have focused on low-level stimuli. Chong and Treisman [5] proposed that observers can concurrently extract average sizes from blue-circle and green-circle sets displayed simultaneously and intermingled in space, respectively. This demonstrated that individuals can categorize a set into different subsets along specific dimensions and extract summary statistics separately. Additionally, Elias and Sweeny [21] displayed within-category or cross-category sets, which comprised several ellipses with distinct aspect ratios and could be classified as tall or flat, then participants were asked to move the mouse to adjust a probe stimulus to match the average aspect ratio of the set (i.e., method-of-adjustment task). They found that participants performed better on the within-category sets than on the cross-category ones. These findings supported the thoughts proposed by Im et al. [22] and Utockkin [23], who suggested that observers form different feature distributions for various category items. In other words, the cross-category sets were perceived as a two-peak distribution for each category (extracting two average representations), rather than a single-peak distribution (extracting a single average representation). Nonetheless, Maule et al. [9] displayed a set of hues chosen from a continuous percept and manipulated the relationship between the members of the sets to be within-category or between-category, with the position of the boundary hue verified by an identification task before the formal experiment. The participants were instructed to judge whether the probe hue was a member of the previously displayed set. The results revealed that participants could extract average information from the between-category sets, and the familiarity of unseen average hues was comparable with that of the actually seen hues. This study provided evidence for ensemble coding in the cross-category sets. In addition to the differences in tasks, the discrepancy between Elias and Sweeny [21] and Maule et al. [9] could be due to set variance. In Experiment 2 of Maule et al.’s study [9], they found that the familiarity effect of unseen average hues disappeared when the perceptual distance between the hues of set members was increased.

However, it is still considerably unclear whether the category relationship of the members of a set with high-level features, such as facial expressions, affects ensemble coding. On the one hand, low-level features may differ from high-level features during the processing of ensemble coding [24]. Therefore, we cannot apply the low-level feature discovery directly to high-level features. On the other hand, a group of faces interacting with us, whether spatially or temporally, are usually heterogeneous in categories in daily life. For example, a group of people may express quite different attitudes towards a particular thing or a person’s expression may change dramatically due to an unexpected event, which results in a crowd whose members express cross-category emotions (i.e., a cross-category crowd presented simultaneously) or a person whose expression changes at different times (i.e., a cross-category crowd presented sequentially). More importantly, as the ability to extract statistical information from a group of emotional faces is crucial to our lives and well-being [12,20], it is necessary to investigate whether perceivers can form averaged representations from a group of facial expressions with different emotional categories.

Here, we aimed to examine whether perceivers could extract average expressions from a set of cross-category facial expressions through two experiments in which facial expressions were presented spatially and temporally, respectively. We focused on happy and fearful expressions rather than happy and angry ones, which are commonly utilized in studies on ensemble coding of facial expressions (e.g., [18,19,20]). This is because happiness and anger are recognized as emotions associated with approach motivation (happiness is related with well-being, while anger is associated with attack). Fear, however, not only belongs to a different emotional category than happiness [25,26,27,28], but also is associated with avoidance motivation (fear relates to withdrawal), located at the opposite end of the motivation dimension from happiness [29,30], which can be categorized more clearly and are commonly used as stimuli in studies on the categorical perception of facial expression [31,32,33]. We chose two faces near the boundary as members of the sets for two reasons. First, in real life, people often express subtle and indistinguishable facial expressions rather than prototypical basic emotions [34,35]. Second, according to previous empirical evidence [9,12,21,22,36,37] and theory [23], as the variance of a set increases, the ability to extract an average representation from a set decreases. A larger variance may prevent us from finding the potential effect of ensemble coding in cross-category crowds. Importantly, it is unusual to encounter a group of people expressing completely different facial expressions concurrently, or a person changing their expression from an extreme category to another extreme category at a glance. Although studies have reported the categorical perception of emotional faces as evidence for the basic theory of emotion [28], ensemble coding of facial expressions supports the dimension theory of emotion [25,38,39], which implies that observers can perceive facial expressions as a continuous percept. This means that observers may extract representations of categorical boundaries and the mean emotion of the cross-category crowds, as in Experiment 1 of Maule et al.’s study [9]. Given the seamless experience of perceiving the diverse world around us in real life, as well as the findings of ensemble coding studies [3,13,17], it is reasonable to assume that observers can extract the average facial expression from a cross-category crowd.

## 2. Experiment 1

In Experiment 1, we simultaneously displayed participants with four images of the same person with different emotions in a member identification task (see below detailed) to investigate whether the participants represented a group of cross-category facial expressions presented simultaneously as an average. Based on the aforementioned studies [3,9,13,17], we hypothesized that participants would perceive a face displaying the average facial expression of a cross-category crowd as a member of that crowd, even if it was never actually presented within the crowd.

### 2.1. Method

***Participants.*** Forty-six undergraduate and graduate students (twenty-eight females; age: *M* = 20.52 years, *SD* = 2.19) with normal or corrected-to-normal vision volunteered to take part in the study and received money for their participation. All of them were right-handed and were included in the analysis. Participants provided written informed consent before the experiment and this study was approved by the ethics committee of Liaoning Normal University (LL2023054). 

According to an a priori statistical power analysis using G*Power [40], this experiment would need a minimum of 13 participants to detect an effect of *f* = 0.25 with 1 − β = 0.95, α = 0.05. 

***Stimuli.*** Two actors, a male and a female, portraying happy and fearful expressions were selected from the NimStim face set [41]. Given that teeth can impact the perception of facial expressions [42,43] and to minimize ghosting effects during morphing [44], all of the faces used in the current study were closed-mouth. These faces were cropped to remove hair, backgrounds, and everything below the chin and were transformed into grayscale using Photoshop 2020 software (version 21.0.1). The two expressions of each actor were morphed with FantaMorph 5 (version 5.4.8) to produce linear continua from happiness to fear. To define the categorical boundary between two emotions, nine faces were chosen from each continuum and were separated by 10% emotional distances (defined by morphing unit, Figure 1a). A new set of 20 participants (10 females; age: *M* = 22.55 years, *SD* = 0.94) were recruited to perform an identification task, in which each face was randomly presented in the centre of the screen for 400 ms and participants were asked to press the ‘j’ key for happy facial expression and press the ‘k’ key for fearful facial expression to determine whether the face expressed happiness or fear. There were 216 trials in total, with each face appearing 12 times. We plotted Figure 1c based on the performances in the identification task and the categorical boundary was defined as the point on the morphing continuum that was equally likely to be categorized as either happiness or fear. The results revealed that the boundary of the male actors was the face containing a 50% happiness and 50% fear mix. The boundary of the female actors was the face with a 55% happiness and 45% fear mix. Six faces around the boundary were selected as the stimuli for the formal experiment for both male and female faces (Figure 1b). As Figure 1b shows, these faces were separated by 18% emotional distances and were coded as two sequences (sequence 1 and 2). Each sequence consisted of five faces, which were coded from A to E, with the proportion of fearful emotion gradually increasing. Each ensemble stimulus used in the formal experiment consisted of four faces of identical identity, which were placed in an invisible 2-row by 2-column grid pattern at the centre of the screen. Each face was assigned to a random position in the matrix at the start of each trial. There were two types of ensemble stimuli: BD ensembles, which comprised two B faces and two D faces, and B/D ensembles, which consisted of either four B faces or four D faces. From this design, the crowds may express one of two types of ensemble emotions: happiness when they consisted of B1 faces and D1 faces or only B faces, and fear when they consisted of B2 faces and D2 faces or only D faces (see Figure 2a). The function of B/D ensembles is to ensure that the effects observed in BD ensembles cannot be attributed to either one of the two faces, B or D, but rather to the ensemble coding for the combination of B faces and D faces. If it is not true, we would observe a similar pattern of response in the B/D conditions as in the BD conditions. All of faces employed in both the identification task and formal experiment were subtended to a 3° × 4° visual angle with the monitor placed at a viewing distance of 70 cm. 

***Procedure.*** Participants were seated approximately 70 cm away from a 19-inch computer monitor with a resolution of 1440 × 900 and a refresh rate of 60 Hz. Each trial began with a fixation for 300 ms, followed by an ensemble stimulus presented at the centre of the screen for 2000 ms while the fixation remained visible. After a blank screen with a fixation was presented in the centre for 1300 ms, a probe face was displayed in the centre. A signal (i.e., words ‘F: no’ and ‘J: yes’) would be presented at the bottom of the screen after the probe face was presented for 400 ms (see Figure 2b) to remind participants to judge whether it is a member of the previous set by pressing ‘j’ for yes and ‘k’ for no. There was 2000 ms for participants to respond, and the probe face would disappear once participants pressed a given key or the time ran out. The probe face could be any one of the five faces from the A face to E face, which belonged to the same sequence and had an identical identity with the previously presented ensemble stimulus. 

There were two blocks in the experiment: one block included faces from sequence 1 and the other included faces from sequence 2. The order of the two types of blocks was counterbalanced between participants. Each block consisted of 90 trials, with 30 BD ensemble trials, 30 B only trials, and 30 D only trials among these trials. 

***Statistical analyses.*** Statistical analyses were conducted using SPSS Statistics for Windows (Version 25.0). Repeated-measures analyses of variance (ANOVAs) were performed for familiarity, which was defined as the proportion of trials that answered ‘yes’, with ensemble type (BD ensemble, B/D ensemble), ensemble emotion (happiness, fear), and probe face (A to E) as within-subjects factors. We did not analyse response times (RTs) because the participants were not required to respond as quickly as possible in the experiment, and there was no explainable effect in RTs. The Greenhouse–Geisser correction was applied when assumptions of sphericity were violated and *p* values less than 0.05 were considered statistically significant unless otherwise specified.

### 2.2. Results

The familiarity of each probe face in different conditions is shown in Figure 3. There were significant main effects of ensemble type (*F*(1, 45) = 220.99, *p* < 0.001, $\eta_{p}^{2}$ = 0.83) and probe face (*F*(4, 180) = 145.31, *p* < 0.001, $\eta_{p}^{2}$ = 0.76) but not ensemble emotion (*F*(1, 45) = 0.035, *p* = 0.853, $\eta_{p}^{2}$ = 0.001). More importantly, a significant three-way interaction among all three factors was observed (*F*(4, 180) = 269.49, *p* < 0.001, $\eta_{p}^{2}$ = 0.86). In the BD ensemble condition, the familiarities of probe faces B (happy ensemble: *M* = 88.77%, *SE* = 2.32; fearful ensemble: *M* = 91.67%, *SE* = 2.06), C (happy ensemble: *M* = 88.77, *SE* = 2.32; fearful ensemble: *M* = 89.13%, *SE* = 2.09), and D (happy ensemble: *M* = 88.04, *SE* = 2.05; fearful ensemble: *M* = 87.32%, *SE* = 2.44) were significantly larger than the familiarities of probe faces A (happy ensemble: *M* = 53.62%, *SE* = 4.54; fearful ensemble: *M* = 59.78%, *SE* = 3.93) and E (happy ensemble: *M* = 40.22%, *SE* = 4.35; fearful ensemble: *M* = 42.39%, *SE* = 4.35) for both happy and fearful ensembles (*p*s < 0.001). No significant differences were found among B, C, and D for both the happy and fearful ensembles (*p*s = 1.000) and between A and E for the happy ensembles (*p* = 0.093), but the familiarity of A was significantly larger than E for the fearful ensemble (*p* < 0.001). In contrast, in the B/D ensemble condition, when the ensemble emotion was happiness, the familiarity of probe face B (*M* = 97.65%, *SE* = 0.89) was significantly larger than that of the other probe faces (*p*s < 0.001); the familiarities of A (*M* = 71.01%, *SE* = 3.60) and C (*M* = 64.31%, *SE* = 3.69) were significantly larger than that of D (*M* = 15.04%, *SE* = 2.72) and E (*M* = 2.36%, *SE* = 0.62, *p*s < 0.001); the familiarity of D was significantly larger than that of E (*p* < 0.001); and no significant differences were found between A and C (*p* = 0.681). When ensemble emotion was fear, the familiarities of probe face D (*M* = 0.96, *SE* = 0.01) was significantly larger than that of the other probe faces (*p*s < 0.001); probe faces C (*M* = 65.76%, *SE* = 3.24) and E (*M* = 58.70%, *SE* = 3.70) were significantly larger than thar of A (*M* =3.99%, *SE* = 1.15) and B (*M* = 16.67%, *SE* = 2.63, *p*s < 0.001); probe face B was significantly larger than that of A (*p* < 0.001); and no significant differences were found between C and E (*p* = 0.546).

To further rule out the possibility of the familiarity effect of probe face C being caused by either B faces or D faces alone, we compared the familiarities of probe face C in the BD ensemble condition to those in the B/D ensemble condition. The results showed that the familiarity of probe face C in the B/D ensemble condition was lower than those in the BD ensemble condition for both happy and fearful ensembles (*p*s < 0.001). The familiarities of probe face C did not differ significantly between the two conditions when the ensemble stimulus was comprised B faces alone or D faces alone (*p* = 0.668).

To eliminate the potential probability that the bias to average facial expression was caused by the familiarity effect formed within the first block, where the average facial expressions were presented (e.g., in the second block, when presenting faces from sequence 1, face C1, which serves as the average facial expression in the BD ensemble condition, would be presented as face B2 within the first block), we conducted a 2 (emotion within the first block: happiness and fear) × 5 (probe face: A to E) mixed ANOVA by using the data of the first block, where the first factor was between-subjects and the second factor was within-subjects. The results demonstrated similar patterns to the main findings described above. Specifically, there was a significant main effect of probe face (*F*(2.91, 128.10) = 75.72, *p* < 0.001, $\eta_{p}^{2}$ = 0.63). The familiarities of probe faces B (*M* = 90.31%, *SE* = 2.05), C (*M* = 87.66%, *SE* = 2.31), and D (*M* = 88.86%, *SE* = 1.94) were significantly higher than that of A (*M* = 58.01%, *SE* = 4.41) and E (*M* = 39.30%, *SE* = 3.63) (*p*s < 0.001), and the familiarity of probe face A was significantly higher than that of E (*p* = 0.002). The differences among probe faces B, C, and D were not significant (*ps* = 1.000). No other effects were found to be significant (*p*s > 0.05). 

### 2.3. Discussion

As mentioned above, in the BD ensemble condition, even though probe face C (the average representation of the previously presented ensemble) was not presented in the ensemble stimuli, it was perceived as equally familiar as the actually presented probe faces (B faces and D faces), and the familiarities of these three probe faces were significantly larger than those of other unpresented probe faces (A faces and E faces). This average bias accords with previous studies regarding ensemble coding of facial expressions in a member judgement task [17,45,46], where observers tend to recognize the unseen average facial expression as a member of previous ensemble stimuli, implying that observers form an average representation when they perceive a crowd of faces automatically. However, these studies either only showed within-category crowds or did not discriminate the contributions of cross-category crowds from within-category crowds. More importantly, our results expand on Maule et al.’s [9] findings by suggesting that observers can extract average representation from a cross-category set, and that the process occurs when high-level stimuli (i.e., facial expressions) are presented simultaneously, not just when low-level stimuli (i.e., hue) are displayed simultaneously. The findings from the B/D ensemble condition exclude the possibility that the high familiarity of probe face C was due to either B or D alone, as no similarity was observed in the patterns of the familiarities of probe face C between the BD ensemble condition and B/D ensemble condition. Considering that the bias towards average facial expressions was also observed within the first block, this cannot be solely attributed to the familiarity effect of average facial expressions formed within that block.

## 3. Experiment 2

In Experiment 1, we found that observers could extract an average facial expression representation from a crowd whose members express cross-category emotions. As in most ensemble coding studies, the crowds used in Experiment 1 were a set of faces with identical identities displayed simultaneously (e.g., [13,47]). However, we rarely meet numerous faces of a person concurrently; instead, we usually interact with a person whose emotions change constantly and even shift between different emotional categories. Previous studies have suggested that ensemble coding also occurs when a sequence of facial expressions is perceived [11,19]. Ensemble coding may underlie the process through which observers perceive a person’s facial expression at the moment of emotional category change. However, these studies also could not provide evidence for ensemble coding of the cross-category crowds in the same way that the previously reported simultaneously presented ensemble could. In Experiment 2, participants were shown a series of faces and required to perform a member judgement task to explore whether observers could extract the average emotional representation from a group of faces presented sequentially. Based on prior studies on ensemble coding, irrespective of whether the stimuli were presented sequentially or simultaneously, similar results were observed [11,17,18,19]; we predicted that we would find resulting patterns similar to those of Experiment 1 in a sequential task.

### 3.1. Method

***Participants.*** The individuals who took part in Experiment 1 participated in this experiment. The participants provided written informed consent before the experiment and this study was approved by the ethics committee of Liaoning Normal University (LL2023054).

***Stimuli and procedure.*** The stimuli and procedure utilized in this experiment were similar to those described in Experiment 1. Different from the 4 faces being presented simultaneously in Experiment 1, 20 faces were displayed sequentially with each face presented for 50 ms followed by an interstimulus interval of 50 ms in Experiment 2. There were also two types of ensemble stimuli: BD ensembles, in which B faces and D faces were displayed 10 times each in random order, and B/D ensembles, in which either B or D faces were presented 20 times (see Figure 4).

***Statistical analysis.*** In addition to the analyses performed in Experiment 1, we compared the familiarity of each probe face between Experiments 1 and 2 to determine whether there was a significant effect of presentation type. We conducted a repeated-measures ANOVA with presentation type (simultaneous, sequential), ensemble type (BD ensemble, B/D ensemble), ensemble emotion (happiness, fear), and probe face (A to E) as within-subjects variables for these two experiments for each participant.

### 3.2. Results

Figure 5 depicts the familiarity of each probe face under the different conditions. There were significant main effects of ensemble type (*F*(1, 45) = 518.21, *p* < 0.001, $\eta_{p}^{2}$ = 0.92) and probe face (*F*(4, 180) = 83.42, *p* < 0.001, $\eta_{p}^{2}$ = 0.65) but not ensemble emotion (*F*(1, 45) = 0.67, *p* = 0.419, $\eta_{p}^{2}$ = 0.02). More importantly, a significant three-way interaction among all three factors was observed (*F*(4, 180) = 230.33, *p* < 0.001, $\eta_{p}^{2}$ = 0.84). In the BD ensemble condition, the familiarities of probe faces B (happy ensemble: *M* = 93.84%, *SE* = 1.82; fearful ensemble: *M* = 88.77%, *SE* = 2.01), C (happy ensemble: *M* = 89.49%, *SE* = 2.71; fearful ensemble: *M* = 87.68, *SE* = 2.40), and D (happy ensemble: *M* = 82.61%, *SE* = 2.79; fearful ensemble: *M* = 86.59%, *SE* = 2.05) were significantly larger than the familiarities of A (happy ensemble: *M* = 58.33%, *SE* = 4.71; fearful ensemble: *M* = 67.03%, *SE* = 4.32) and E (happy ensemble: *M* = 48.91%, *SE* = 4.32; fearful ensemble: *M* = 56.88%, *SE* = 4.16) for both happy (D-A: *p* = 0.001; B-D: *p* = 0.007; others: *p*s < 0.001) and fearful ensembles (C-A: *p* = 0.002; D-A: *p* = 0.001; others: *p* < 0.001). No significant differences were found among probe faces B, C, and D, or between faces A and E (*p*s > 0.05). In contrast, in the B/D ensemble condition, when the ensemble emotion was happiness, the familiarity of probe face B (*M* = 94.75%, *SE* = 1.20, *p*s < 0.001) was significantly larger than that of the other probe faces; probe faces A (*M* = 68.12%, *SE* = 3.48) and C (*M* = 58.70%, *SE* = 3.25) were significantly larger (*p*s < 0.001) than that of D (*M* = 10.33%, *SE* = 1.99) and E (*M* = 2.54%, *SE* = 0.77); the familiarity of D was significantly larger than that of E (*p* = 0.004); and no significant differences were found between A and C (*p* = 0.115). When the ensemble emotion was fear, the familiarity of probe face D (*M* = 93.84%, *SE* = 1.05) was significantly larger than that of the other probe faces (*p*s < 0.001); the familiarities of probe faces C (*M* = 60.51%, *SE* = 3.16) and E (*M* = 56.34%, *SE* = 3.64) were significantly larger (*p*s < 0.001) than that of A (*M* = 3.80%, *SE* = 1.18) and B (*M* = 13.77%, *SE* = 2.28); the familiarity of probe face B was significantly larger than that of A (*p* < 0.001); and no significant differences were found between C and E (*p* = 1.000).

To rule out the possibility that the familiarity effect of probe face C was induced solely by B or D faces, we compared the familiarities of probe face C in the BD ensemble condition to those in the B/D ensemble condition. The findings revealed that the familiarities of probe face C were lower in the B/D ensemble condition than in the BD ensemble condition for both happy and fearful ensembles (*p* < 0.001). The familiarities of probe face C were not significantly different between the two conditions where the ensemble stimuli were B faces alone or D faces alone (*p* = 0.565).

To eliminate the probability that the bias to average facial expression was caused by the familiarity effect formed within the first block, where the average facial expressions were presented, we conducted a 2 (emotion within the first block: happiness and fear) × 5 (probe face: A to E) mixed ANOVA by using the data collected from the first block, where the first factor was between-subjects and the second factor was within-subjects. The resulting patterns were similar to the main findings described above. Specifically, there was a significant main effect of probe face (*F* (2.64, 116.18) = 27.39, *p* < 0.001, $\eta_{p}^{2}$ = 0.38). The familiarities of probe faces B (*M* = 90.40%, *SE* = 1.98), C (*M* = 88.67%, *SE* = 2.38) and D (*M* = 85.92%, *SE* = 2.56) were significantly higher than that of A (*M* = 63.16%, *SE* = 2.58) and E (*M* = 54.58%, *SE* = 4.10) (C vs. A: *p* = 0.001; D vs. A: *p* = 0.002; other: *p*s < 0.001), and the differences among probe faces B, C, and D were not significant (*p*s = 1.000). No other effects were significant (*p*s > 0.05).

Comparing the familiarities of probe face C between Experiments 1 and 2, we found that there were no significant main effects of presentation type (*F*(1, 4) = 0.01, *p* = 0.935, $\eta_{p}^{2}$ < 0.001) or interaction among the four factors (*F*(4, 180) = 1.05, *p* = 0.376, $\eta_{p}^{2}$ = 0.02).

However, the absence of a comparison condition (i.e., within-category condition) in Experiments 1 and 2 may prevent us from fully understanding the ensemble coding of cross-category crowds. After Experiments 1 and 2, we conducted two further experiments corresponding to them, with the only difference between them being the inclusion of a within-category condition. In the further experiments, the performance of a group of 16 participants (who did not participate in the formal experiment but did participate in both further experiments) replicated the average bias observed in Experiments 1 and 2 under the cross-category condition, suggesting that the average bias also occurred in the within-category condition. Specifically, the familiarity of the unseen average face was comparable to that of the actually presented faces (all *p*s > 0.05), but both were higher than other unseen faces (all *p*s < 0.05). Comparing the familiarities of average faces between the within-category and cross-category conditions, we found a similar level of familiarity of average faces between the two conditions (all *p*s > 0.05) in both happy and fearful crowds.

### 3.3. Discussion

The results of Experiment 2 were the same as those of Experiment 1, which suggested that the unseen probe face C as an average representation of the ensemble stimulus would be as familiar as those that were actually presented in the ensemble stimulus (i.e., probe faces B and D) in the BD ensemble condition. This effect could not be explained by either faces B or D alone because when ensemble stimuli comprised only faces B or D, the familiarity of probe face C decreased. This finding cannot be solely attributed to the familiarity effect of average facial expressions formed within the first block, as it was observed even when analysing the data of the first block. This again revealed that the observers were able to extract an average representation while perceiving a group of cross-category facial expressions, even if these faces expressed different emotions. The results of this experiment expanded the effect of cross-category ensemble coding from spatial to temporal. Consistent with Kramer et al.’s [48] findings, we did not find a potential effect of the presentation type of facial crowds on ensemble coding. These findings suggested that observers were able to utilize a single representation, specifically the average facial expression, to effectively represent both a cross-category crowd and an individual’s continually changing facial expression and the performance of this average bias for the cross-category crowds was similar to that of the within-category crowds, as evidenced by our supplementary experiments.

## 4. General Discussion

Through two experiments, this study demonstrated that observers can extract average representations from a set of cross-category facial expressions whether they are presented simultaneously or sequentially. The participants preferred to recognize the unpresented average facial expression of a group of cross-category facial expressions as one of its constituents. Interestingly, the ensemble coding was so powerful that this preference equalled the preference for the actually presented faces. To strengthen the persuasiveness of our conclusion, we excluded three potential factors that could have influenced the results. First, this average bias could not be attributed to any single face of the sets, face B or D, because the effect disappeared when the sets were composed of only one of these two faces, and the rates of recognizing the average face as a member of the set in the B/D condition were lower than in the BD condition. Second, this effect could not be solely attributed to the familiarity effect caused by the members of ensemble stimuli in the first block serving as the average facial expressions in the second block. Finally, the bias towards average facial expressions observed in the BD ensemble condition could not be exclusively explained by the fact that face C was similar to both the B and D faces, while the other unseen faces (i.e., A and E) were similar to only one face. This is because the familiarity of C face is not only higher than that of unseen faces but it is also similar to the faces that were actually seen. To fully understand the nature of ensemble coding of cross-category crowds, we conducted two further experiments corresponding to Experiments 1 and 2, which added a within-category condition as a comparison condition. The results revealed a similar extent of average bias in the within-category and cross-category conditions, which further confirmed our main findings in Experiments 1 and 2 (i.e., observers can extract average representation from a cross-category crowd), demonstrating that the ensemble coding of cross-category crowds may be similar to that for within-category ones.

These findings are consistent with previous results that used the member judgement task [17,45,46], which found that observers preferred to recognize the unseen average facial expression as a member of the previously presented set. However, some studies suggest that the integration of a group of facial expressions as an average representation using ensemble coding depends on the shape of the feature distribution [22,23], that is, observers perceive a group of cross-category facial expressions as two subsets and form two average representations separately because these crowds are perceived as two-peak distributions. Additionally, several studies on the linkage between mean and variance in ensemble coding demonstrated that observers’ ability to form an average is impaired with an increase in set variance [9,21,37,49,50]. Observers regard the cross-category sets as more heterogeneous than the within-category sets because the performance of discrimination between stimuli that transcend the categorical boundary was better than that for those that did not cross the boundary, evidence of categorical perception [28]. For these reasons, it is possible that a group of cross-category facial expressions cannot be integrated as a single representation by ensemble coding, potentially because averaging is not an ideal representative method for a heterogeneous set [9], which contradicts our results. According to Maule et al.’s study [9], the cross-category average may depend on the perceptual distance between the members. The sets used here and in Maule et al.’s Experiment 1 [9] were constructed of stimuli that are near categorical boundaries in the morphed continuum from one category to another, unlike the sets employed in other studies that contradict our results, which comprised items with large perceptual distances (e.g., [21,49]). 

There are two potential ways to form a single average representation from a group of cross-category facial expressions when the perceptual distances in the sets are sufficiently short. Observers may build an average that is not dependent on the emotional category as in basic theory [25,26,27,28], but rather by emotional valence, which is a continuous dimension in dimension theory [28,29,30], or by surface facial features alone. On the one hand, the phenomenon of forming an average representation for a group of facial expressions itself implies that observers perceive a crowd of facial expressions comprising faces located within a morph sequence as a continuum, and the intermediate morphs can be perceived as neutral [28] rather than as discrete categories. On the other hand, although prior studies argued that ensemble coding of facial expressions is a high-level phenomenon [3,12,17,24,44,51,52,53], their findings could not rule out the possibility that low-level features are involved in ensemble coding because they only showed that performance decreased rather than vanished when the faces were inverted. It is well known that both holistic and configural processing, as well as feature-based processing, contribute to single facial expression recognition [54,55,56]. Moreover, previous methods to measure the ensemble coding of facial expressions (e.g., [13,17,18]) usually asked participants to match a probe face to the average of a set, in which participants could make an accurate response by comparing surface facial features between the group face and the probe face, such as the extent of the mouth curve for happy expressions and the extent of frowning for angry expressions. Further investigations are needed to determine whether dimensional information and feature-based information contribute to the cross-category ensemble coding of facial expressions. Nonetheless, regardless of the information used by the observers, comparing the current study with previous studies [9,21,23] suggest that there may be different mechanisms for cross-category ensemble coding based on perceptual distances between items along a given dimension. Specifically, as perceptual distances decrease, the impact of dimensional or feature-based information in cross-category ensemble coding may increase, and categorical information becomes more important as perceptual distances increase. This separation of mechanisms in ensemble coding is beneficial for survival. For instance, we can perceive yellowish fruits or blooms from a cluster of greenish leaves presented together with them on the same tree [23,57], but when leaves turn from green to yellow with the change of seasons, people will not perceive leaves with different colours around the boundary between green and yellow as two subsets. This can help us avoid consuming our cognitive resources.

A few caveats need to be discussed. First, because the actors used in the present study were white, there may be a potential race effect on the results. Nevertheless, this may not have a substantial impact on the results for the critical categorical boundary that was defined by the Chinese participants. However, we cannot completely rule out the racial effect, as we did not specifically test it in our study. Further study is needed to investigate the impact of race on ensemble coding of cross-category facial expressions. Second, there was only one identity among every set presented here, like most previous studies on ensemble coding [13,17,19,58] to avoid the effect of irrelevant factors (e.g., the difference of facial attraction among crowd members). However, new work examining whether current findings can be expanded to a crowd composed of members with different identities can enhance the ecological validity of the existing findings since we always interact with a crowd with different identities in daily life. Third, as the current study only focused on average facial expression, it is still unknown whether our findings can be applied to other summary statistics extracted from the cross-category crowds, such as variance and the shape of the distribution of a group of facial expressions. Accordingly, the effect of category relationship among members on the ability to extract other summary statistics from crowds needs further investigation. Fourth, we did not control the gender ratios in our study. As many studies have suggested the effect of gender differences on emotion perception (e.g., [59,60,61,62,63,64]), it is important to investigate the role of gender on ensemble coding of cross-category crowds in the future studies. 

## 5. Conclusions

This study demonstrated that observers can integrate a group of cross-category facial expressions with short perceptual distances using ensemble coding. This may explain the rapid and accurate perception of heterogeneous groups in the real world. Depending on the perceptual distances between items, ensemble coding of a group of cross-category facial expressions may involve two different mechanisms. This study investigated cross-category ensemble coding of high-level stimuli, which contributes to the understanding of ensemble coding and provides inspiration for future studies.

## Figures and Tables

**Figure 1 behavsci-14-00508-f001:**
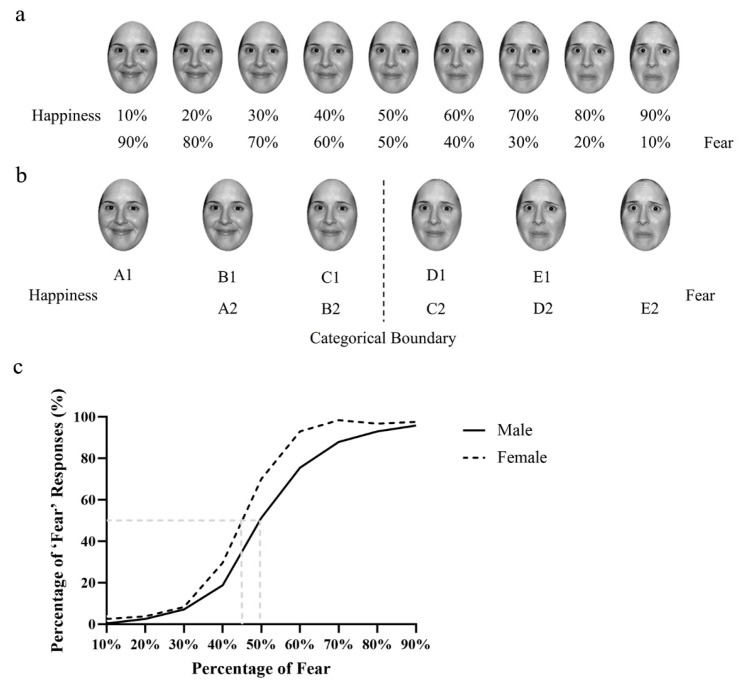
(**a**) The continuum of morphed facial expressions used in the identification task. (**b**) The six faces used in the formal experiments around the categorical boundary, with each face having a morphed distance of 18% from one another; they belong to two sequences, where sequence 1 included the A1, B1, C1, D1, and E1 faces and sequence 2 included the A2, B2, C2, D2, and E2 faces. (**c**) Plot of the results of the identification task. The percentage of ‘fear’ responses in the two-alternative forced choice task, where the results of the male faces are indicated by the black solid line and the results of the female faces are indicated by the black dashed line; it was plotted against the percentage of fearful expression in the morphed continuum. Gray dashed line indicates the position where observers had an equal chance to perceive the faces as either happy or fearful.

**Figure 2 behavsci-14-00508-f002:**
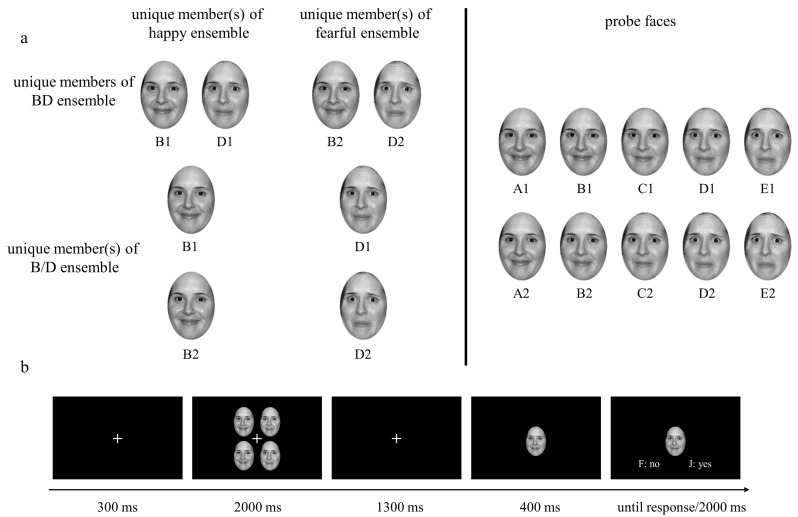
Illustration of unique members of the ensembles in different conditions (**a**) and overview of the procedure of Experiment 1 (**b**). (**a**) The left part shows unique members of ensembles in different conditions. The number of each unique member in the ensembles was as follows: two B1 faces and two D1 faces in the happy BD ensembles, two B2 faces and two D2 faces in the fearful BD ensembles, four B1 faces or four B2 faces in the happy B/D ensembles, and four D1 faces or four D2 faces in the fearful B/D ensembles; the right section displays all the probe faces. In a single trial, the probe face was one of the five faces of each sequence, which was consistent with the ensemble stimuli. (**b**) Each trial began with a fixation displayed at the centre of the screen for 300 ms, followed by an ensemble stimulus for 2000 ms. Then, a probe face was presented at the centre of the screen following a blank screen displayed for 1300 ms. A signal (i.e., words ‘F: no’ and ‘J: yes’) reminding participants to respond was presented at the bottom of the screen after the probe face was presented for 400 ms. The probe face disappeared once the participants pressed a given key or after the signal was displayed for 2000 ms.

**Figure 3 behavsci-14-00508-f003:**
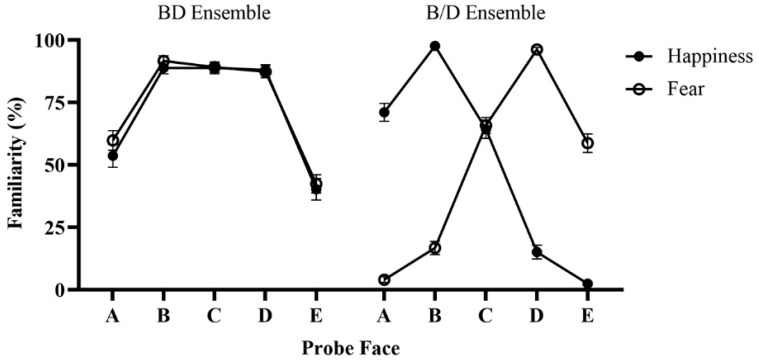
The results of Experiment 1. Error bars represent standard errors.

**Figure 4 behavsci-14-00508-f004:**
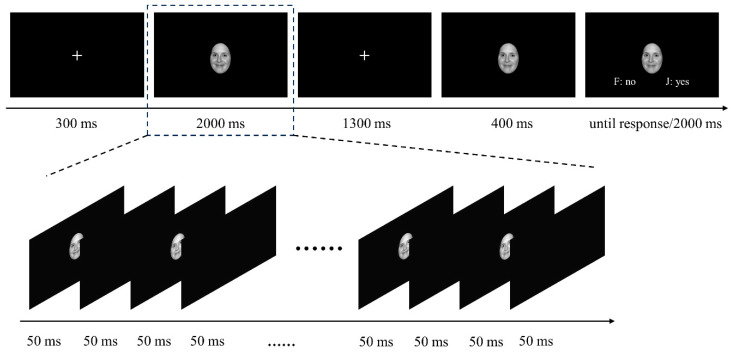
Overview of the procedure of Experiment 2. Each trial began with a fixation displayed at the centre of the screen for 300 ms, followed by an ensemble stimulus consisting of 20 faces presented sequentially for 2000 ms, with each face appearing for 50 ms after a 50 ms blank screen. Then, a probe face appeared at the centre of the screen following a blank screen being displayed for 1300 ms. After the probe face appeared for 400 ms, a signal (i.e., words ‘F: no’ and ‘J: yes’) reminding participants to respond appeared at the bottom of the screen. The probe face disappeared once the participants pressed a given key or after the signal was displayed for 2000 ms.

**Figure 5 behavsci-14-00508-f005:**
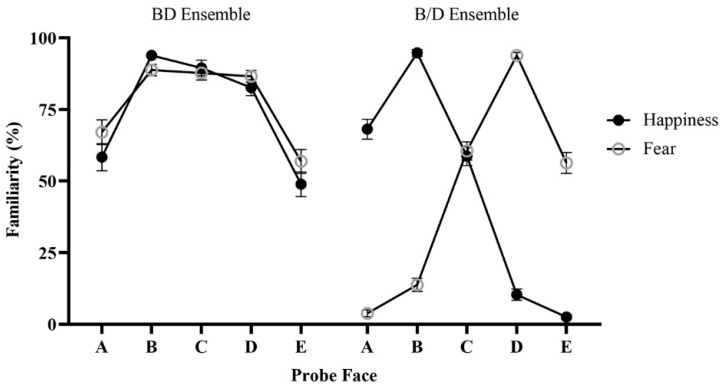
The results of Experiment 2. Error bars represent standard errors.

## Data Availability

Data will be made available on request.

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
