# Peer review of "Ensemble Coding of Crowd with Cross-Category Facial Expressions"

_behavsci, 2024, doi:10.3390/bs14060508_

Round 1
Reviewer 1 Report
Comments and Suggestions for Authors
The manuscript “Ensemble coding of crowd with cross-category facial expressions” by Yang and co-authors studied representation of facial expression from a group of stimuli. The group of stimuli were either spatially or temporally distributed. The authors found that participants tend to classify an averaged facial expression that they have not seen before as a member of the dominant expression of the group.
My major concern about the manuscript is that the authors’ description of their rationale and significance was not written in a clear manner, and as a result, the novelty of the study was not obvious to me.
1. From the introduction, I was not able to understand what are the issues that previous studies of ensemble coding of facial expressions failed to solve, and the authors managed to provide a solution to.
2. Line 110-112. The authors claim that they used happy and fearful faces instead of happy and angry ones because happy and fearful faces have different emotional categories and are opposite in the motivational dimension. However, to me, happy and angry faces also belong to different emotional categories and are quite different in the motivational dimension. Thus, this reasoning does not seem to strongly support the authors’ choice of happy and fearful faces in the study.
3. Line 186-193. The description about the design of BD and B/D ensembles are quite confusing. The authors should show example figures for these two conditions to better describe their experimental design.
4. There is no description about the relationship between the gender of the subjects and the gender of the stimuli used. This is important as participants may have different performance in recognition of facial expression from the same or the opposite gender.
5. Line 266, “we” is missing before “compared”.
6. In the discussion session of the first experiment, the authors claimed that their findings replicate a previous study. And then in the discussion session of the second experiment, the authors claimed that their second experiment replicates the result of the first one. Thus, it seems to me that all the results the authors found are mostly supporting theories/findings of previous studies.
Comments on the Quality of English LanguageThe writing definitely needs to be improved. The authors could clarify and emphasize their originality compared to previous studies, their experimental design, and their theoretical contributions.
Author Response
Please see the attachment about the response letter.

Reviewer 2 Report
Comments and Suggestions for Authors
In two experiments, the current study examined whether observers can extract an average facial expression from a cross-category set. This was done by measuring participants' tendency to identify a face as a member of the crowd. It was demonstrated that an average face was judged as familiar, even though it was not presented, in a cross-category set. This was evident when the crowd was presented as a group and also in a successive presentation. The theoretical question is of great value, yet I have several concerns regarding the experimental procedure.
1) If I understood correctly (if not, maybe a clearer description of the method is required), the variance in faces presented was larger in the BD conditions compared to the B/D conditions (in which only B or D were presented). This might make it difficult to compare the results between the two conditions. It seems that similar variance in facial expressions should be employed in all conditions for valid comparison between them.
2) Relatedly to the first comment, one experimental condition that I think is missing from the study is a condition that examines the likelihood of perceiving an average face that was not presented as part of the crowd, in a within category assembly. Will that face also be recognized similarly to faces that were actually presented? Maybe it will be perceived as more familiar than presented faces?
Author Response

(The authors gave the same response as above.)

Round 2
Reviewer 1 Report
Comments and Suggestions for Authors
My comments have been satisfactorily addressed by the authors.
Author Response
Many thanks for your helpful comments.Reviewer 2 Report
Comments and Suggestions for Authors
I thank the authors for providing their responses to my comments. I still think that having smaller variance in the B/D condition compared to the BD condition might bias the results. It is possible that the larger variance of the BD condition produces an overestimation of the possibility that a given face belongs to the ensemble. A better comparison in my view would be to compare between ensembles with similar variance in the within- and between-category conditions.
This would also address my second comment, which will enable a direct comparison within and between category conditions.
My comments require the authors to run an additional experiment. I leave it to the editor to decide whether these comments should prevent publication of the MS in its current form or not.
Author Response
Please see the attachment about the response letter. Thank you.

Round 3
Reviewer 2 Report
Comments and Suggestions for Authors
I still think that the suggested experiment is important in order to have a valid control condition for comparison. I leave it to the editor to decide if without such a condition the paper is suitable for publication in its current form.
Author Response
Please see the attachment for the response letter.
